# Displacement, Strain and Failure Estimation for Multi-Material Structure Using the Displacement-Strain Transformation Matrix

**DOI:** 10.3390/ma13010190

**Published:** 2020-01-02

**Authors:** Hye-Lim Jang, Dae-Hyun Han, Mun-Young Hwang, Donghoon Kang, Lae-Hyong Kang

**Affiliations:** 1Department of Mechatronics Engineering, and LANL-JBNU Engineering Institute-Korea, Jeonbuk National University, 567 Baekje-daero, Deokjin-gu, Jeonju-si, Jeollabuk-do 54896, Korea; janghr7320@jbnu.ac.kr (H.-L.J.); dh.han@jbnu.ac.kr (D.-H.H.); munyoung.h@jbnu.ac.kr (M.-Y.H.); 2Railroad Safety Research Team, Korea Railroad Research Institute, 176, Railroad Museum St, Uiwang, Gyeonggi-do 16105, Korea; 3Department of Flexible and Printable Electronics, Department of Mechatronics Engineering, and LANL-JBNU Engineering Institute-Korea, 567 Baekje-daero, Deokjin-gu, Jeonju-si, Jeollabuk-do 54896, Korea

**Keywords:** displacement-strain transformation, deformation, strain, failure, structural health monitoring

## Abstract

In this study, we propose a method to estimate structural deformation and failure by using displacement-strain transformation matrices, i.e., strain-to-displacement transformation (SDT) and displacement-to-strain transformation (DST). The proposed SDT method can be used to estimate the complete structural deformation where it is not possible to apply deformation measurement sensors, and the DST method can be used for to estimate structural failures where strain and stress sensors cannot be applied. We applied the SDT matrix to a 1D beam, a 2D plate, rotating structures and real wind turbine blades, and successfully estimated the deformation in the structures. However, certain difficulties were encountered while estimating the displacement of brittle material such as an alumina beam. The study aims at estimating the displacement and stress to predict the failure of the structure. We also explored applying the method to multi-material structures such as a two-beam bonded structure. In the study, we used alumina–aluminum bonded structures because alumina is bonded to the substrate to protect the structure from heat in many cases. Finally, we present the results of the displacement and failure estimation for the alumina–aluminum structure.

## 1. Introduction

It is necessary to consider maximum deformation, strain, stress and failure of the structure to ensure safe operations. Deformation and strain can be measured using displacement sensors, such as linear variable differential transformers (LVDTs) [1], potentiometers [2], laser displacement sensors [3], strain sensors, such as strain gages [4] and fiber optic sensors [5]. Generally, stress and failure can be estimated based on the strain information. However, in some cases, it is not possible to measure the deformation by using displacement sensors. If we imagine an aircraft in operation, it is difficult to apply displacement sensors because displacement sensors should be installed on a fixed frame. In other cases, it is not possible to measure strain by using strain sensors. For example, strain gages cannot be used under very high electromagnetic and thermal environments or severe vibration. Alternatively, it is possible to use strain-based displacement estimation and displacement-based strain estimation methods.

Many studies have focused on structural deformation estimation based on strain information [6,7,8,9,10], and these studies are summarized in Table 1. As shown in Table 1, strain measurement was performed via strain gages or fiber optic sensors that measure multiple strains along a single strand of the fiber. The maximum estimation error ranges from 1.5%–7%. Researchers used displacement-strain transformation techniques [6,7,10], developed multimetric data fusion techniques [8] and used a linear classical beam-curvature function with a fiber optic strain sensing algorithm (FOSS) [9]. 

However, strain or stress estimation methods using displacement data were not commonly used, because strain sensors can be easily used for many applications, and it is occasionally difficult for displacement sensors to measure actual strains due to slip or failure to measure the reference point. However, if it is not possible to use strain sensors in high temperature conditions in an oven, then displacement data can be alternatively used by laser displacement sensors to estimate the strain or stress of the structure [11]. It is also possible to apply the image registration [12] or modal superposition method [13] for strain estimation.

The proposed displacement or strain estimation method in the study is based on a modal approach. The use of classical beam–curvature functions increases time consumption due to the increase in structural complexity. However, the modal parameter-based approach can reduce the estimation calculation time and be applied in vibration control applications. Furthermore, the proposed method exhibits a significant advantage wherein the complete structural shape or stress information can be calculated by using a small number of sensors. Figure 1 illustrates the process of estimating the structural deformation or strain by using the displacement–strain transformation matrix. 

The authors’ group is continuously developing and improving the displacement–strain transformation method, and the research in our group is summarized in Figure 2. The displacement–strain transformation method has been actively studied since the early 2000s. At the beginning of the study, deformation predictions for one-dimensional beam shapes [14] and two-dimensional plates were conducted [15]. After that, the deformation predictions were obtained by gradually expanding its application range to rotating structures [16] and actual wind turbine blades [17]. 

The present study focuses on multi-material structures such as two-beam bonded structures. This is because many bonded structures are used for certain applications. For example, the application of multi-material structures, such as composite–metal and ceramic–metal combinations, is increasing [18,19,20]. These types of composite structures suffer from problems including delamination and de-bonding. If we apply displacement–strain transformation techniques, then we can obtain the structural deformation and strain information as well as the failure and de-bonding information in a non-destructive manner.

In the study, we first applied the displacement–strain transformation method to a single material structure to verify strain estimation by using displacement data, as well as the deformation estimation by using strain data. Subsequently, a multi-material structure was used in the same verification test. In addition to the simple prediction of deformation or strain, we propose a method that can determine the point of failure and presence of damage by calculating the stress induced in the structure when deformation occurs. The DST method has been widely applied to single materials, but has not been applied to multi-material or brittle materials. In this paper, we wanted to verify the feasibility of deformation prediction by using the DST method to multi-material and brittle structures.

## 2. Background Theory of the Displacement-Strain Transformation

### 2.1. Relationship between Displacement and Strain

The displacement is predicted by measuring strain because the displacement of the structure is not independent of the strain if the structure corresponds to a continuum. The displacement and strain are expressed as the product of the mode matrix, and the modal coordinates are expressed as follows:(1)$$\left\{d\right\}=\left[\phi_{N}\right]\left\{\eta_{N}\right\}$$
(2)$$\left\{s\right\}=\left[\psi_{N}\right]\left\{\eta_{N}\right\}$$
where, *N* denotes the number of modes used, {*d*} denotes the displacement, {*s*} denotes the strain, $\left\{\eta_{N}\right\}$ denotes the modal coordinate vector, $\left\{\phi_{N}\right\}$ denotes the displacement mode matrix, and $\left\{\psi_{N}\right\}$ denotes the strain mode matrix. If we rearrange {*d*} and {*s*} with the modal coordinate matrix, we obtain Equations (4) and (5):(3)$$\left\{\eta_{N}\right\}={\left({\left[\psi_{N}\right]}^{T}\left[\psi_{N}\right]\right)}^{-1}{\left[\psi_{N}\right]}^{T}\left\{s\right\}$$
(4)$$\left\{\eta_{N}\right\}={\left({\left[\phi_{N}\right]}^{T}\left[\phi_{N}\right]\right)}^{-1}{\left[\phi_{N}\right]}^{T}\left\{d\right\}$$
where, ${\left[\psi_{N}\right]}^{T}$ denotes the transposed matrix of $\left[\psi_{N}\right]$, and we substitute Equations (3) and (4) into Equations (1) and (2) to obtain Equations (5) and (6) as follows:(5)$$\left\{d\right\}=\left[\phi_{N}\right]{\left({\left[\psi_{N}\right]}^{T}\left[\psi_{N}\right]\right)}^{-1}{\left[\psi_{N}\right]}^{T}\left\{s\right\}$$
(6)$$\left\{s\right\}=\left[\psi_{N}\right]{\left({\left[\phi_{N}\right]}^{T}\left[\phi_{N}\right]\right)}^{-1}{\left[\phi_{N}\right]}^{T}\left\{d\right\}$$

Here, $\left[\phi_{N}\right]{\left({\left[\psi_{N}\right]}^{T}\left[\psi_{N}\right]\right)}^{-1}{\left[\psi_{N}\right]}^{T}$ and $\left[\psi_{N}\right]{\left({\left[\psi_{N}\right]}^{T}\left[\psi_{N}\right]\right)}^{-1}{\left[\psi_{N}\right]}^{T}$ are expressed as strain-to-displacement transformation (SDT) and displacement-to-strain transformation (DST), respectively. Finally, the displacement and the strain are expressed as Equations (7) and (8), respectively:{*d*} = [SDT]{*s*}(7)
{*s*} = [DST]{*d*}(8)

In the above equation, it is observed that the displacement and strain at all positions of the structure are obtained by the mode shape matrix and the measured {*s*} and {*d*} [21].

### 2.2. Number of Strain Gage Sensors

The accuracy of estimation increases when the number of sensors increases in the estimation of the structural deformation. However, it is difficult to use several sensors, given the cost of sensors and the difficulty of attachment and wiring to the structure. Thus, structural deformation can be estimated more effectively if the sensors are optimized for quantity and location.

Based on the rank of the displacement–strain transformation matrix, the number of sensors is identical to the order of mode shapes of the structure. It is possible to determine the optimal number of sensors if we determine the primary mode shape of the structure. In this study, the cantilever beam is used to estimate the structural deformation, and the loading is applied on the free edge. Therefore, the primary mode shape of the beam should correspond to the first mode of the bending mode shape. Hence, the number of sensors corresponds to one or more than one. In this study, three strain sensors are used for a more accurate strain prediction. 

### 2.3. Optimization for Sensor Location

Given the limited number of sensors that can be used, the selection of an optimum measuring position allows for more accurate deformation predictions. In the study, the location of the strain sensor is determined via a genetic algorithm. The genetic algorithm is an imitation method of the law of survival of the fittest and crossing over [22]. In the implementation of the genetic algorithm for optimization, the population size, the number of subpopulations and the maximum number of generations for evolution must be chosen [23]. Figure 3 presents the flow chart of the genetic algorithm. The population is selected to ensure an optimal solution. A new population is created via the cloning and mating of this population. After determining the solution wherein the conditions are satisfied, this sequence is repeated until the optimal solution is determined. In this study, the optimal solution is obtained via minimizing condition number (CN) [24]. CN is the relationship between the minimum value and the maximum value in the matrix. In the case of the DST matrix, it was determined that the smaller the CN value showed a higher accuracy of the prediction.

## 3. Estimation of Deformation and Strain of a Single Material Structure

In the experiment, a stainless steel beam (300 mm × 25 mm × 1 mm) was used. The material properties of the stainless steel are listed in Table 2. The strains were measured by strain gages attached to a Wheatstone bridge circuit, and the displacements were measured via laser displacement sensors.

### 3.1. Sensor Location Optimization

The optimal location for attaching the strain gage sensors to stainless steel specimens was selected via MATLAB’s optimtool genetic algorithm. Considering the repeated process of the genetic algorithm, it was confirmed that the optimum position of the three strain gage sensors corresponded to 39, 48 and 159 mm. We used the same approach and obtained the optimal displacement sensor locations for three points measurements as 96, 196 and 287.

### 3.2. Estimation of Deformation of a Single Material Structure Using the SDT Method

#### 3.2.1. Experiment Setup for Deformation of Single Material Structure

The structure was deformed by applying loads corresponding to 50 gf, 100 gf and 150 gf to the end tip of the stainless steel beam, as shown in Figure 4. The deformation of the stainless steel beam was estimated via the proposed SDT matrix and measured strain values. The locations of the strain sensors for structural deformation estimation was selected as 39, 48 and 159 mm via the genetic algorithm. Strain gages were attached to these three locations, as shown in Figure 5 and the strains were obtained. The displacement of the stainless steel beam was measured via a laser displacement sensor, and the measurement point was measured at the endpoint of the beam.

#### 3.2.2. Displacement Estimation Results Using SDT Method

To validate the estimation accuracy of the SDT and DST methods, experiment results were compared with the results of the SDT and DST methods, finite elements method (FEM) and the analytical method. Specifically, ANSYS was used for the FEM simulation. In the analytical method, to obtain the deflection of the beam, Equations (9) and (10) are used as follows:(9)$$\delta_{max}=\frac{Pl^{3}}{3EI}$$
(10)$$I=\frac{bh^{3}}{12}$$
where $\delta_{max}$ denotes the maximum deflection of the beam, P denotes the force at the endpoint of the beam, l denotes the length of the beam, E denotes the elastic modulus of the beam, I denotes the moment of inertia in the rectangular cross-section of the beam, b denotes the width of the cross-section of the beam, and h denotes the height of the cross-section of the beam.

Figure 6 shows the results obtained by applying the measured strain values to the SDT matrix and comparing the displacements that were calculated by using the predicted displacement and beam deflection equations and the displacement values obtained by the analytical method employing ANSYS.

The results of the displacement prediction indicated that when the specimen was subjected to loads corresponding to 50, 100 and 150 gf, the displacement value appeared as a curve with the largest displacement at the end of the beam. Additionally, it was observed that the experimental values, values predicted by using the SDT matrix, and displacement values calculated by the analytical method, agree well with each other.

In order to determine the accuracy of the displacement prediction using the SDT matrix, error analysis was performed for each prediction and experiment result. Table 3 shows the error rates of displacement values obtained via analytical methods, experimental values and predicted displacement values using the SDT matrix.

The error was less than 5.2% between experiment value and estimation value using the SDT matrix. As shown in Table 3, the results of the SDT matrix are significantly accurate when compared with those of the FEM and analytical methods.

### 3.3. Estimation of the Deformation in a Single Material Structure by Using the DST Method

#### 3.3.1. Experimental Setup for Deformation Estimation of a Single Material Structure

The locations of the displacement sensors were selected as corresponding to 96, 196 and 287 mm from the genetic algorithm. Two laser displacement sensors measured the structural deformation at 96 mm and 196 mm. The deformation at 287 mm is measured using a ruler, as shown in Figure 7. At the 287 mm point of the beam, the structure was deformed by pulling the beam by 10, 20, and 30 mm. With respect to the 96 mm and 196 mm locations, as shown in Figure 8 and the displacement of each point was obtained via laser displacement sensors. The strain of the stainless steel beam was measured with a strain indicator, and the measurement position was identical to the position of the strain gage sensor in the SDT experiment.

#### 3.3.2. Strain Estimation Results Using the DST Method

As shown in Figure 9, the experimental values, predicted values using the DST matrix and strain values calculated by the analytical method agree well with each other. In order to calculate the accuracy of the strain prediction method using the DST matrix, error analysis was performed for each prediction and experiment result. Table 4 shows the results of strain estimation. The error in the DST method is less than 7.9%. 

## 4. Estimation of Deformation and Strain of the Multi-Material Structure

The displacement–strain transformation technique was also applied to a multi-material structure. In the study, the multi-material structure was composed of an aluminum beam, alumina beam and epoxy bonding layer. Table 5 lists the material properties of each component of the multi-material structure.

The dimensions of the alumina and aluminum beam corresponded to 100 mm in length, 25 mm in width and 1 mm in thickness, as shown in Figure 10. The thickness of the epoxy layer was measured as 0.15 mm after the bonding of two beams. The epoxy bonding layer was composed of a 2:1 weight ratio of epoxy resin (PRO-SET ADV-175) and hardener (PRO-SET ADV-275) and cured at room temperature for 10 h. The length of the specimen was determined to be shorter than that of the stainless steel beam that was used in single material structure presented in Section 3 because the cost of the alumina ceramic is very high, and we examined when a rupture of the ceramic occurred during the test. The specimen was fixed using a clamp, and thus, the dimensions of the actual cantilever beam corresponded to 62 mm in length, 25 mm in width, and 2.15 mm in total thickness.

### 4.1. Sensor Location Optimization

The location of the strain gage sensor was optimized via the genetic algorithm and CN value. Thus, the strain sensor location was determined as 11, 13 and 37 mm, and displacement sensor location was calculated as corresponding to 20, 37 and 56 mm. However, it is not possible to set the positions of the strain gage sensors to the optimized locations due to the difficulty in attaching the strain gage sensor because it was extremely close between the optimized positions. The position of strain gage sensors was determined as 9, 13 and 33 mm for the purpose of convenience. In the case of the laser displacement sensor, the length of the beam was short and installation of the laser displacement sensors was difficult due to the head size. The positions of the laser displacement sensors were set at 22 mm and 52 mm to avoid overlapping of the laser head. Although the position of the sensor was set in a manner different from the optimized position, it could be sufficiently applied to the SDT and DST matrix. Given that the modified sensor position and optimized sensor position were similar to each other, it was not considered to significantly affect the result.

### 4.2. Experiment Setup for Deformation Estimation of Multi-Material Structure

Figure 11 shows the complete experimental setup for the deformation and strain estimation of the multilayer structure. An end of the specimen was fixed and the other end was deformed in the out-of-plane direction via the linear stage until the specimen was broken. Displacement and strain of the multilayer structure were measured via laser displacement sensors and strain gages, respectively.

### 4.3. Displacement Estimation Results Using the SDT Method

Figure 12 and Table 6 show the displacement estimation results using SDT and measured displacements at 22 mm and 52 mm on the alumina side of the beam. The multilayer beam was broken when the deformation of the endpoint of the beam approximately corresponded to 1.2 mm. The estimation error between the experimental value and estimation value using SDT was less than 16.2%.

### 4.4. Strain Estimation Results Using the DST Method

Figure 13 and Table 7 present the strains estimated using DST and the strain measured at 9, 13 and 33 mm on the alumina side of the beam. As shown in Figure 13, the experimental and predicted values for the DST matrix match well with each other. Table 7 shows the results of the strain estimation, and the error in the DST method is less than 7.6%.

### 4.5. Stress and Failure Estimation Results Using the DST Method

The stress estimation was conducted via the DST method. Figure 14 and Table 8 summarize the result of stress estimation. The multilayer beam was deformed until the beam was broken, and the stress of the multilayer beam was calculated by multiplying the strain and estimated from DST and elastic modulus of the alumina. In the experiment, the multilayer beam was broken when the deformation at the endpoint of the multilayer beam exceeded approximately 1.2 mm. When the deformation of the beam corresponded to 1.2 mm, it was confirmed that the stress applied to the beam is more than 200 MPa. Given that the yield strength of alumina was approximately in the range of 215–248 MPa when the elastic modulus was approximately 330–370 GPa [25], it was observed that the failure occurred because the stress generated by the deformation applied to the alumina beam exceeded the yield strength of the alumina.

## 5. Conclusions

In the study, the structural deformation and failure estimation methods using DST and SDT were proposed. We verified the displacement-strain transformation method for the structural deformation and strain estimation of single- and multi-material structures.

First, the DST and SDT methods were applied to a single material structure to verify the strain estimation using displacement data and deformation estimation using strain data. A comparison of the deformation and strain values of the single structure confirmed that the experimental values, predicted values using the DST and SDT matrix and displacement and strain values calculated by the analytical method, agreed well each other.

Second, we verified the multi-material structure. The DST and SDT method was applied to an aluminum–alumina beam. A comparison of the displacement and strain of the multilayer beam indicated that the experimental values and predicted values using the DST and SDT matrix matched well with each other.

Finally, we performed the stress and failure estimation of the multilayer beam. The failure estimation of the multi-material structure was conducted via the DST matrix. The failure prediction results confirmed that the estimated stress using the DST matrix was similar to the stress obtained using the ANSYS simulation. It was also observed that the stress at beam failure was well-matched with the yield strength of the alumina.

It was concluded the displacement–strain transformation method can be used for the deformation and strain estimation of the structure where the displacement or strain sensor cannot be applied. In addition to the simple prediction of the displacement or strain, DST and SDT methods determine the point of failure and presence of damage by calculating the stress of the structure when deformation occurs. The proposed method uses a smaller number of discrete alternative sensors to estimate complete structural deformation, strain and stress distribution. Therefore, it can be used for real-time structural behavior monitoring or control without the need for intensive calculations and numerous sensors.

## Figures and Tables

**Figure 1 materials-13-00190-f001:**
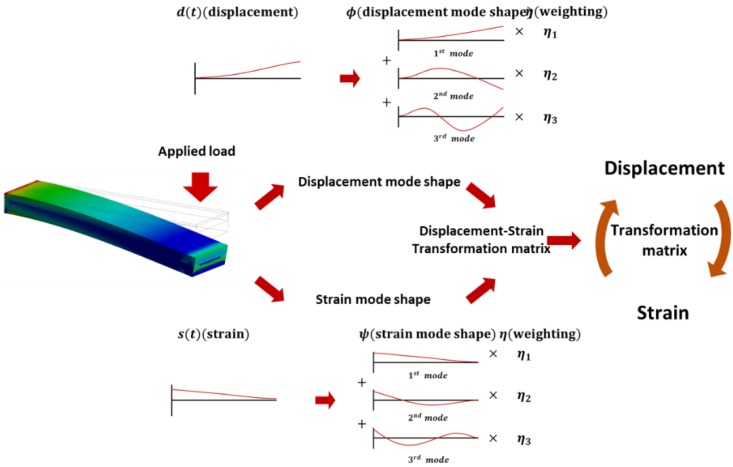
Process of estimating the structural deformation or strain.

**Figure 2 materials-13-00190-f002:**
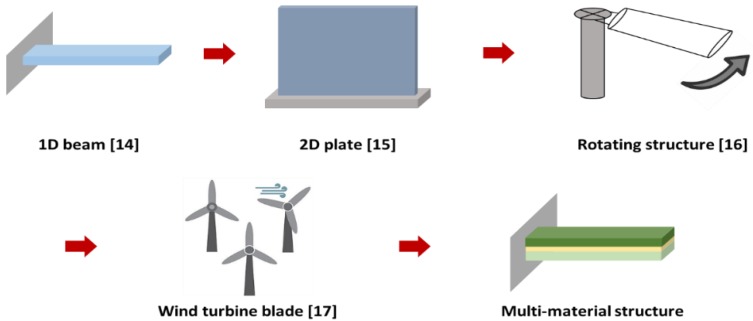
Research flow of the displacement–strain transformation method used in the study.

**Figure 3 materials-13-00190-f003:**
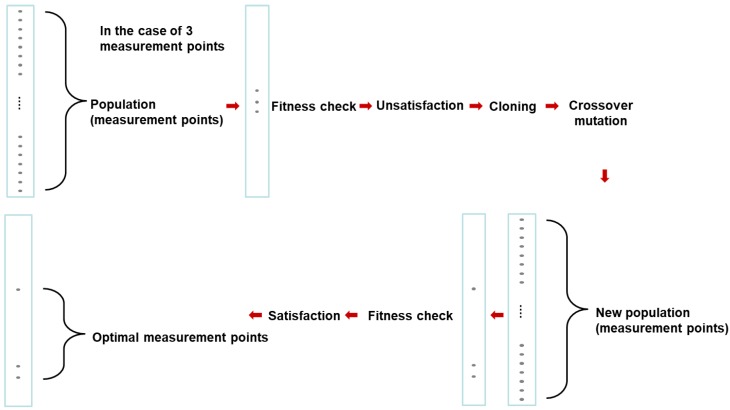
Optimization of the strain gage sensor location.

**Figure 4 materials-13-00190-f004:**
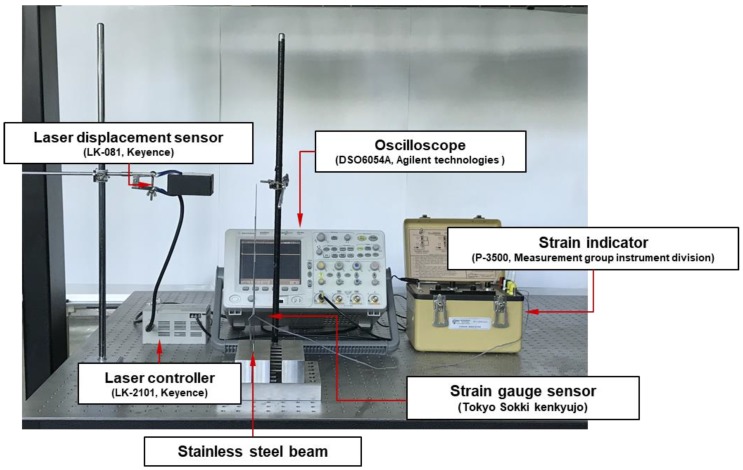
Test setup for strain-to-displacement transformation experiment.

**Figure 5 materials-13-00190-f005:**
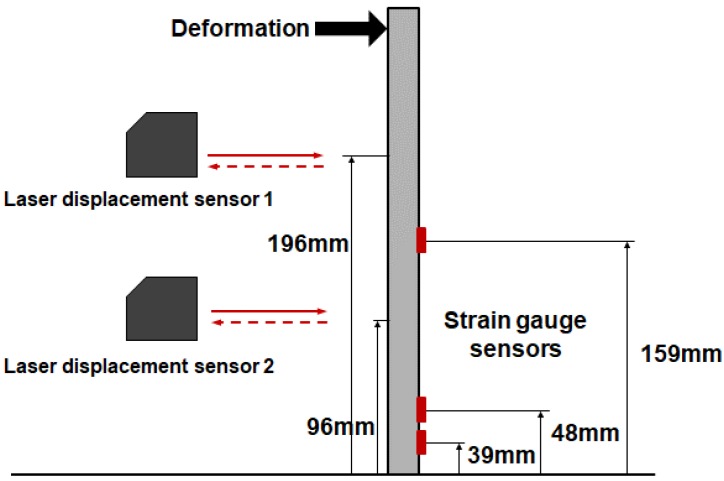
Measurement points of the stainless steel beam.

**Figure 6 materials-13-00190-f006:**
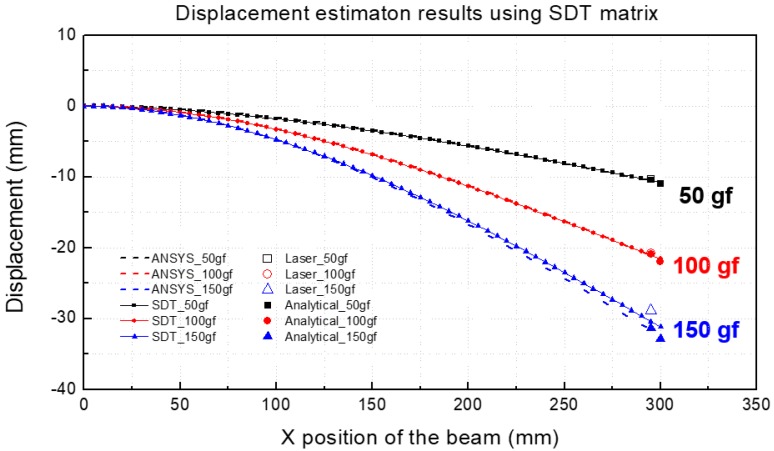
Displacement estimation results by using the strain-to-displacement transformation (SDT) matrix (Single material).

**Figure 7 materials-13-00190-f007:**
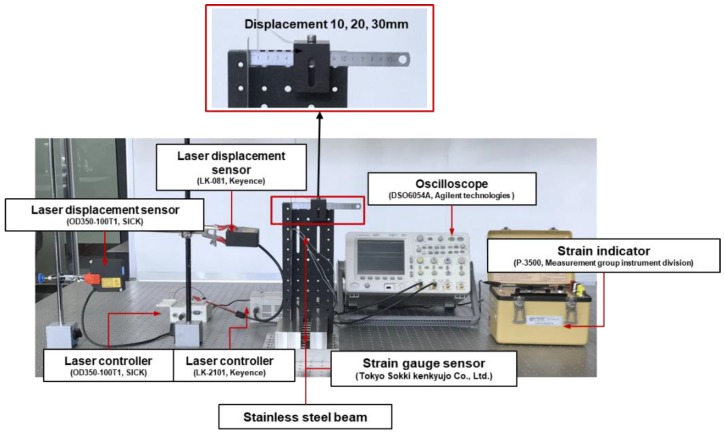
Test setup for displacement-to-strain transformation experiment.

**Figure 8 materials-13-00190-f008:**
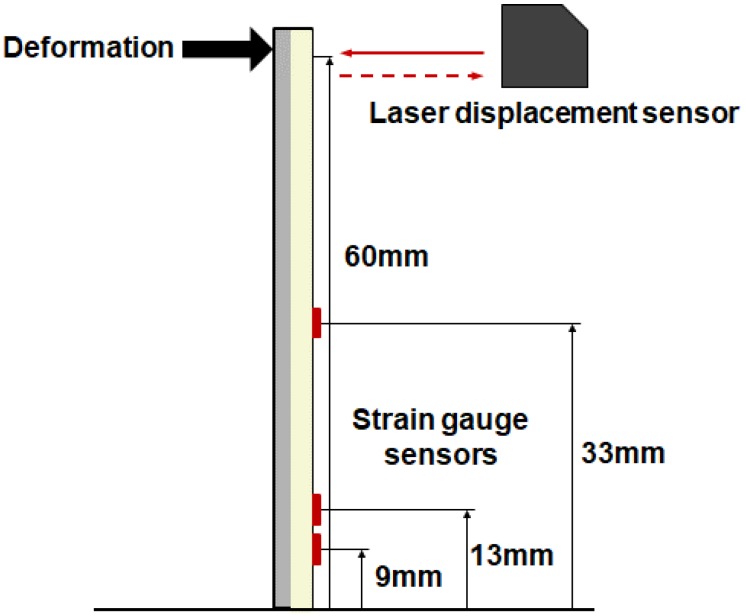
Measurement points of the multilayer beam.

**Figure 9 materials-13-00190-f009:**
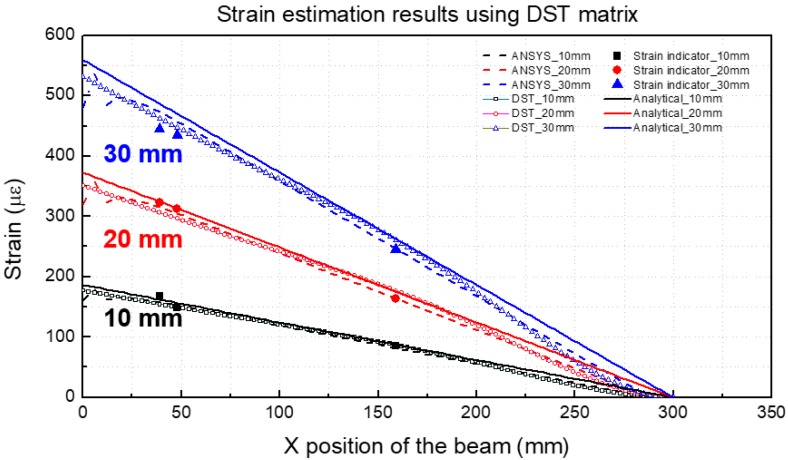
Strain estimation results using the DST matrix (Single material).

**Figure 10 materials-13-00190-f010:**
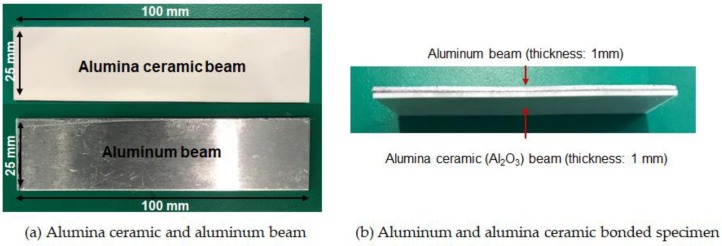
Aluminum and alumina ceramic bonded specimen.

**Figure 11 materials-13-00190-f011:**
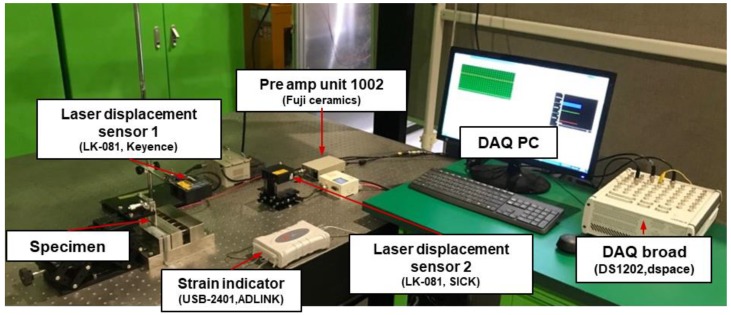
Test setup for deformation and strain estimation experiment for the multilayer beam.

**Figure 12 materials-13-00190-f012:**
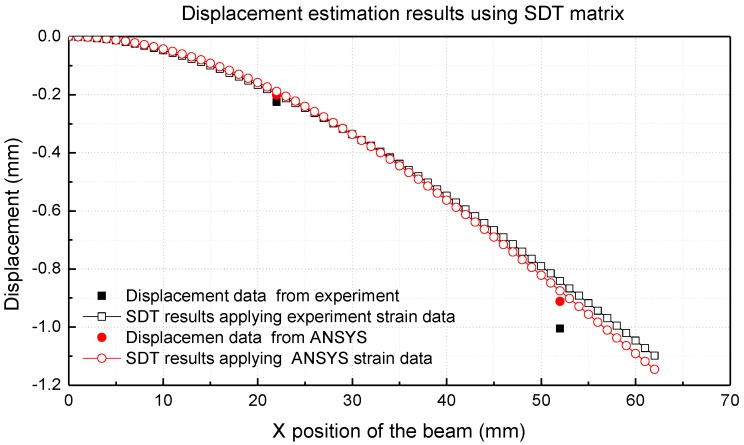
Displacement estimation results using the SDT matrix (multi-material).

**Figure 13 materials-13-00190-f013:**
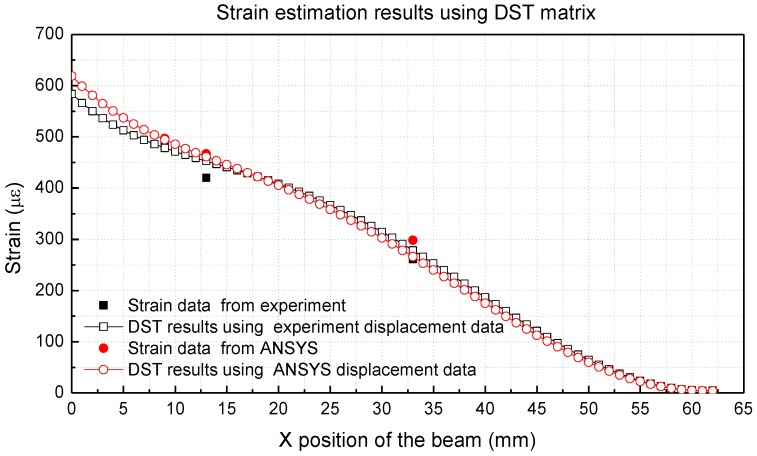
Strain estimation results using the DST matrix (multi-material).

**Figure 14 materials-13-00190-f014:**
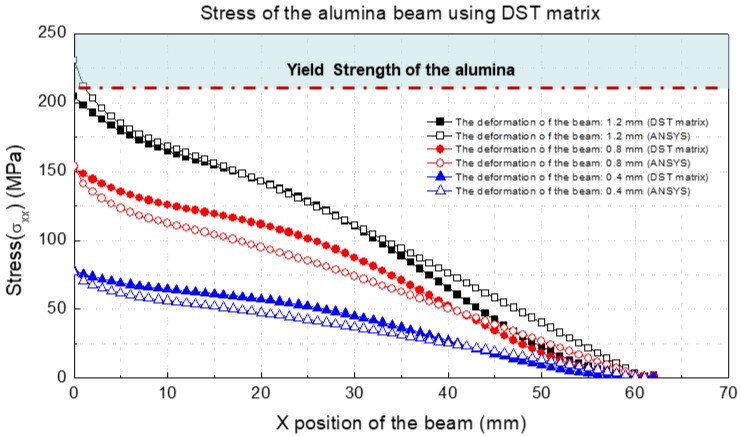
Stress of the alumina beam using the DST matrix.

**Table 1 materials-13-00190-t001:** Comparison of the deformation estimation method.

Previous Studies	Structure	Estimation Method	Estimation Value	Sensing Method	No. of Sensors	Loading Condition	Accuracy
[6]	Cantilever plate	DST matrix	Displacement	FBG Sensor	16	Shaker excitation	Max error: 2.2%
[7]	Wind turbine tower	DST matrix	Displacement	FBG Sensor	10	Operating turbine	Max error: 1%
[8]	Beam structure	Multimetric data fusion	Displacement	FBG Sensor	15	Moving load (0.1 m/s)	Max error: 7%
[9]	Plate	FOSS algorithm	Displacement	FBG Sensor	100	Load	Max error: 3.7% (0.57 cm)
[10]	Beam structure	DST matrix	Displacement	Strain gage sensor	7	Hammering	Max error: 7%
[11]	SMC lead	DST matrix	Strain	Accelerometer	35	Vibration	Max error: 1%
[12]	3D tendon	Image registration	Strain	Ultrasound image	1	Axial load	Max error: 1.5%
[13]	Cantilever beam	Video motion estimation and modal superposition	Strain	Strain gage sensor	5	Vibration	MAC: 0.997

**Table 2 materials-13-00190-t002:** Material properties of the stainless steel beam.

Material Properties	Unit	Value
Density	kg/m^3^	7750
Elastic modulus	GPa	193
Poisson’s ratio	-	0.3

**Table 3 materials-13-00190-t003:** Displacement estimation results using the SDT matrix (Single material).

Load (gf)	Location of the Laser Displacement Sensor (mm)	Displacement (mm)			
		Experiment	ANSYS	SDT	Analytical Method
50	295	−10.31	−10.54 (2.2%)	−10.47 (1.5%)	−10.43 (1.1%)
100	295	−20.82	−21.07 (1.2%)	−21.02 (0.9%)	−20.86 (0.1%)
150	295	−28.86	−31.60 (9.4%)	−30.38 (5.2%)	−31.29 (8.3%)

( ): Displacement estimation error calculated based on the experimental value.

**Table 4 materials-13-00190-t004:** Strain estimation results using the DST matrix (Single material).

Deformation of the Beam at 287 mm	Location of the Strain Gage Sensor (mm)	Strain (με)			
		Experiment	ANSYS	DST	Analytical Method
10 mm	39	167	157.63 (5.6%)	154.30 (7.6%)	162.28 (2.8%)
	48	149	152.40 (2.3%)	149.20 (0.1%)	156.68 (5.2%)
	159	85	82.06 (3.5%)	86.73 (2.0%)	87.66 (3.1%)
20 mm	39	323	315.26 (2.4%)	306.39 (5.1%)	324.56 (0.5%)
	48	312	304.80 (2.3%)	296.32 (5.0%)	313.36 (0.4%)
	159	164	164.14 (0.1%)	176.89 (7.9%)	175.33 (6.9%)
30 mm	39	445	472.90 (6.3%)	462.57 (3.9%)	486.84 (9.4%)
	48	435	457.20 (5.1%)	447.00 (2.8%)	470.05 (8.1%)
	159	245	246.20 (0.5%)	260.94 (6.5%)	263.00 (7.3%)

( ): Strain estimation error calculated based on the experimental values.

**Table 5 materials-13-00190-t005:** Material properties of the aluminum and alumina ceramic.

Properties	Unit	Aluminum	Alumina Ceramic	Epoxy
Density	kg/m^3^	2270	3900	1100
Modulus of elasticity	GPa	71	350	2.9
Poisson’s ratio	-	0.33	0.23	-
Tensile yield strength	MPa	280	220	50
Compressive yield strength	MPa	280	2900	82

**Table 6 materials-13-00190-t006:** Displacement estimation results using the SDT matrix (multi-material).

Location of the Laser Displacement Sensor (mm)	Displacement (mm)			
	Experiment	SDT Results from Strain Data on Experiment	ANSYS	SDT Results from Strain Data on ANSYS
22	−0.224	−0.196 (−12.5%)	−0.201 (−10.2%)	−0.188 (−16.0%)
52	−1.004	−0.841 (−16.2%)	−0.912 (−9.1%)	−0.875 (−12.8%)

( ): Displacement estimation error calculated based on the experimental value.

**Table 7 materials-13-00190-t007:** Strain estimation results using the DST matrix (multi-material).

Location of the Strain Gage Sensor (mm)	Strain (με)			
	Experiment	DST Results from Strain Data on Experiment	ANSYS	DST Results from Strain Data on ANSYS
9	492	478 (−2.8%)	497 (1.0%)	494 (0.4%)
13	420	452 (7.6%)	467 (11.1%)	461 (9.7%)
33	261	278 (6.5%)	298 (14.1%)	265 (1.5%)

( ): Strain estimation error calculated based on the experimental value.

**Table 8 materials-13-00190-t008:** Stress of the alumina beam using the DST matrix.

Deformation at the Endpoint (mm)	Maximum Stress on the Alumina Beam (MPa)		Error (%)
	DST	ANSYS	
0.4	77	76	1.2
0.8	152	153	0.6
1.2	204	230	12.7

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
