# Peer review of "Displacement, Strain and Failure Estimation for Multi-Material Structure Using the Displacement-Strain Transformation Matrix"

_materials, 2020, doi:10.3390/ma13010190_

Round 1
Reviewer 1 Report
The topic of the paper does not seem to fit in the scope of the journal very much. If the highlight of this article is the displacement-strain transformation then it is not innovative at all. The problem statement of this study is not clear so as the purpose of this study. The paper did convey some preliminary heuristic knowledge about the relationship between strain and displacement but this knowledge seems not the original work of the authors. If the condition number and genetic algorithm is the highlight of this work then apparently, they are not elaborated and emphasized enough. This paper also has serious writing issues, for example, there are many repetitions in section 3. Lastly many conclusions of the paper drawn by the authors are very unconvincing. For instance, how come “DST is a reasonable technique to predict strain of the structure” considering such big errors in Table 9. However, it is undeniable that the authors did a great job in describing the experimental setup. The experiments are carried out in a very rigorous way so that all the data are proved to be credible.
Author Response
Please see the attachment.
Thank you for giving us the opportunity to strengthen our manuscript with your valuable comments and queries. We have worked hard to incorporate your feedback and hope that these revisions persuade you to accept our submission.

Reviewer 2 Report
The reviewed paper is original and presents interesting research on estimation of the structural deformation, which is definitely an important issue in all large scale structures, especially those with not typical form, new structural materials and those subjected to complex loads.
Research bases on 2 types of small samples, which in the opinion of the reviewer seems to be rather as pilot tests. Moreover, the samples made out of two different materials differ with dimensions (300mm vs 100mm), which is perhaps not crucial for the aim of the research but rises questions about the clearness, coherence and reliability of the tests and their results.
I would be grateful for the explanation why the Authors of the research decided for two types of geometries and how it influences the results.
Authors name the samples as “beams”, which name in structural engineering is restricted to other type of structural element.
The state of art in the Introduction may be found as very general and referring to too few researches – only 10. Aside of that some of cited papers as [3] does not give the same format of the information (missing max. error, which should be important for the reader), so perhaps should not be used.
The references in the text are quoted with use of the capital letter in the last name of leading author and then ones first name (first 6), which is unnecessary.
References (since v. 404) are written incorrectly (most of them), in at least 3 different ways. Reference 8 is additionally wrong by misunderstanding the names of authors, should be: Pyzara A.; Bylina B.; Bylina J. (…).
This chapter should be ordered and corrected.
The number of elements tested was little and description of the results – very general, proving usability of proposed methods for SHM (which has been proved in numerous publications). Methodology of the research, its description and conclusions are with flaws making this interesting research not fully clear and helpful to other researchers who would like to base on it and develop it further.
Author Response

(The authors gave the same response as above.)

Reviewer 3 Report
The paper is good in-parts, very interesting as well. But, there is something missing, I feel you need to improve especially the following
Font sizes of some figures need to be improved: The introduction is too brief, almost, without crediting fellow researchers, especially those who worked in this area (displacement sensing) optimal location was determined by the genetic algorithm but how the number of such strain gauges is decided? random? will the error depend on the number of sensors/ gages used? section 3 describes a bit of it, but is it just a speculation? can you cite a few articles here to show such relationships, if possible? the latest citations are missing? especially in the last 5 years. need more citations section 2.2 is well explained, any citation here? how many loading can be validated?, have you tried different dimensions of specimens?
Author Response

(The authors gave the same response as above.)

Reviewer 4 Report
This paper proposes to estimate deformations of structures using strain measurement or estimate strains of structures using deformation measurement. My main concern is that the paper does not have sufficient contribution, because the method seems already established and applied in more complicated structures. Perhaps the authors can clarify in a revised version.
Following are some detailed questions and comments that might be useful for improving the quality:
(1) The title and abstract do not reflect the main work accurately. Please consider to revise. The title emphasizes estimation of deformation, but the main body focuses on both estimations of deformation and strain, almost half and half for each. The abstract is misleading as well. The first few sentences are not closely related to the main research.
(2) I am not clear about the technical merits of this paper. The conversion relationship between deformation and strain seems straightforward, and has been studied in quite a few studies and applied in different structures.
(3) In your introduction, it is good to see that you are aware of the advanced measurement technologies, such as fiber optic sensors. However, you should be aware of recent advances in measurement based on fully-distributed fiber optic sensors based on light scatterings. The distributed sensors have been applied to: (i) measure continuous strains in https://doi.org/10.1016/j.matlet.2015.01.140 (ii) detect cracks in https://doi.org/10.1088/1361-665X/aa71f4 and (iii) high-temperature applications in https://doi.org/10.3390/s19040877.
(4) According to Fig. 2, I thought this study mainly focuses on multi-material structure, because the single-material structures have been fully studies in the past. However, there is a whole section on the single-material beam. What is the objective of the section?
(5) In section 2.3, the introduction on the genetic algorithm is too blur. More accurate descriptions should be used.
(6) In section 3, a lot of important information is inaccurate or missing. Here are some examples:
The specimen is only 1 mm so it should be called plate rather than a beam.
Where did you install the strain gauges on the specimen? Please provide a photo.
What strain gauges did you use? Does the mass of the strain gauges affect the vibration characteristics of the specimen?
How about the accuracy of the strain gauges and laser measurement?
How about the measurement frequency?
(7) The modal analysis is based on the vibration of structures. Is it applicable to only dynamic behaviors analysis or applicable for both dynamic and static behaviors analysis?
(8) You simply presented the test results. More in-depth analysis should be added to enhance the quality of the study.
Author Response

(The authors gave the same response as above.)

Round 2
Reviewer 1 Report
Significant improvements can be found throughout the manuscript. The purpose of the study is clearer with the support of references. However, this paper is still lack of innovations comparing with other references listed in Table 1. Clearly, DST matrix is not the authors' own creation and the loading conditions in this work are too common and simple in contrast to other works. Even though the number of sensors is reduced to only 3, it is not imperative at all and actually if more sensors can improve the accuracy, then the priority should be the latter. In addition, Part 2.3 explaining the optimization of sensor locations is very confusing, examples need to be added to prove the selection process. Lastly, the max error is still too large to rely upon and cannot compete against the accuracies of the Refs[6]-[13]. In some cases, it is obviously doubtful that what if the load is kept increasing, will the error become larger and larger? More data points need to be provided to validate the results and methodology.
Besides, again the topic does not seem to fit well with the scope of Materials, other OA journals under MDPI, such as Applied Sciences might be a better match for the field of this study.
Author Response

(The authors gave the same response as above.)

Reviewer 2 Report
Thank you for the comments to the review and proposed corrections. I postulate to accept the paper in corrected form.
Author Response

(The authors gave the same response as above.)

Reviewer 4 Report
Thanks for preparing the replies and answer my questions. With your answers, I have better understandings of your work. I appreciate your effects in putting together the manuscript and replies.
But, I do not see sufficient technical merits to justify the publication on this journal. There are already a lot of studies on the applied method. In fact, there are already many real world applications. I do not see new knowledge or advancement of technology from this paper, so I cannot recommend it in the current form.
However, if it might be interesting and valuable if you present real-world applications with significant findings.